# Genetic Variation Underpinning ADHD Risk in a Caribbean Community

**DOI:** 10.3390/cells8080907

**Published:** 2019-08-16

**Authors:** Pedro J. Puentes-Rozo, Johan E. Acosta-López, Martha L. Cervantes-Henríquez, Martha L. Martínez-Banfi, Elsy Mejia-Segura, Manuel Sánchez-Rojas, Marco E. Anaya-Romero, Antonio Acosta-Hoyos, Guisselle A. García-Llinás, Claudio A. Mastronardi, David A. Pineda, F. Xavier Castellanos, Mauricio Arcos-Burgos, Jorge I. Vélez

**Affiliations:** 1Grupo de Neurociencias del Caribe, Unidad de Neurociencias Cognitivas, Universidad Simón Bolívar, Barranquilla 080002, Colombia; 2Grupo de Neurociencias del Caribe, Universidad del Atlántico, Barranquilla 080007, Colombia; 3División de Ingenierías, Universidad del Norte, Barranquilla 081007, Colombia; 4Grupo de Investigación en Genética, Laboratorio de Genética y Biología Molecular, Universidad Simón Bolívar, Barranquilla 080002, Colombia; 5INPAC Research Group, Fundación Universitaria Sanitas, Bogotá 110211, Colombia; 6Neuroscience Research Group, University of Antioquia, Medellín 050010, Colombia; 7Neuropsychology and Conduct Research Group, University of San Buenaventura, Medellín 050010, Colombia; 8Department of Child and Adolescent Psychiatry, Hassenfeld Children’s Hospital at NYU Langone, New York, NY 10016, USA; 9Nathan Kline Institute for Psychiatric Research, Orangeburg, NY 10962, USA; 10Grupo de Investigación en Psiquiatría (GIPSI), Departamento de Psiquiatría, Instituto de Investigaciones Médicas, Facultad de Medicina, Universidad de Antioquia, Medellín 050010, Colombia

**Keywords:** ADHD, *ADGRL3*, *LPHN3*, *SNAP25*, *FGF1*, genetics, Caribbean community, FBAT, predictive genomics

## Abstract

Attention Deficit Hyperactivity Disorder (ADHD) is a highly heritable and prevalent neurodevelopmental disorder that frequently persists into adulthood. Strong evidence from genetic studies indicates that single nucleotide polymorphisms (SNPs) harboured in the *ADGRL3* (*LPHN3*), *SNAP25*, *FGF1*, *DRD4*, and *SLC6A2* genes are associated with ADHD. We genotyped 26 SNPs harboured in genes previously reported to be associated with ADHD and evaluated their potential association in 386 individuals belonging to 113 nuclear families from a Caribbean community in Barranquilla, Colombia, using family-based association tests. SNPs rs362990-*SNAP25* (T allele; *p* = 2.46 × 10^−4^), rs2282794-*FGF1* (A allele; *p* = 1.33 × 10^−2^), rs2122642-*ADGRL3* (C allele, *p* = 3.5 × 10^−2^), and *ADGRL3* haplotype CCC (markers rs1565902-rs10001410-rs2122642, OR = 1.74, *P*_permuted_ = 0.021) were significantly associated with ADHD. Our results confirm the susceptibility to ADHD conferred by *SNAP25*, *FGF1*, and *ADGRL3* variants in a community with a significant African American component, and provide evidence supporting the existence of specific patterns of genetic stratification underpinning the susceptibility to ADHD. Knowledge of population genetics is crucial to define risk and predict susceptibility to disease.

## 1. Introduction

Attention deficit/hyperactivity disorder (ADHD) is the most common neurodevelopmental behavioural disorder that affects ~5% (with figures reaching 17%) of children and adolescents of different cohorts worldwide [1,2,3,4,5,6]. Although in some patients this condition tends to resolve by adulthood, in other cases it can persist and can have serious life-long health and socio-economic adverse consequences [7]. Affected individuals are at increased risk of poor educational achievement, low-income, underemployment, legal difficulties, and impaired social relationships [8,9]. ADHD increases the risk for disruptive (externalizing) symptoms of conduct disorder (CD), oppositional defiant disorder (ODD), and substance use disorder (SUD) [8,9].

Genetic factors are strongly implicated in the aetiology of ADHD, CD, ODD, and SUD [6,10,11,12]. In particular, common single nucleotide polymorphisms (SNPs) harboured in the Adhesion G-protein-coupled receptor L3 (*ADGRL3*, also known as Latrophilin 3 or *LPHN3*; markers rs2345039, rs6551665, and rs1947274), the Synaptosomal-associated protein of molecular weight 25 kDa (*SNAP25*), the Fibroblast growth factor 1 (*FGF1*), the Solute carrier family 6 (neurotransmitter transporter, noradrenalin) member 2 (*SLC6A2*), and the Dopamine receptor D4 (*DRD4*) genes predispose one to ADHD [6,13], as confirmed by worldwide replications [2,3,6,14,15,16,17,18,19,20,21].

In this study, we explored the association of ADHD with SNPs within these genes in a family-based sample of 386 individuals ascertained from a community inhabiting the city of Barranquilla located in the Caribbean coast of Colombia. Barranquilla, with a population of ~2.4 million, is a cosmopolitan city that represents the confluence of many populations (e.g., aboriginal Amerindian, African, and a complex admixture of European (Spain), Syrian–Lebanese, Sephardic Jew, German, Italian, and English communities). These communities settled in the Atlantic coast of Colombia during the last five centuries [22] and established a pattern of genetic flow [23] that is very different from other communities in Colombia, i.e., the Paisa community [24,25] or the Andean communities surrounding Bogota [26,27]. It has been determined that the Caribbean community has one of the largest African American admixture in Colombia and in general in the Central and South American regions [28].

Our overarching goal was to evaluate the effect of genomic variants, already associated with ADHD, in the susceptibility to develop this condition in families ascertained from this Caribbean community and compare those with genomic variants predictors of ADHD in other Colombian communities. This will allow us to define specific biological predictors and study the role of genetic flow as a pivotal factor shaping the susceptibility to this neuropsychiatric disease.

## 2. Subjects and Methods

### 2.1. Subjects

We studied 386 individuals (218 (56.5%) males and 168 (43.5%) females) from 113 nuclear families whose members were born in the Barranquilla, Colombia, metropolitan area (Table 1). A total of 224 (58%) individuals were diagnosed with ADHD and 162 (42%) were diagnosed as unaffected; 94 (26.2%) were children (6–11 years), 34 (9.4%) adolescents (12–17 years) and 232 (64.4%) adults (>18 years). No children or adults were under medication. The average family size was 3.4 ± 0.65 members, with 74 (65.4%) trios, 33 (29.2%) families with four members, 4 (3.5%) with five members, and 2 (1.8%) with six members [29]. Families belonged to the medium socio-economic stratum, with an average monthly income of ~US$1000–3000. All individuals participated voluntarily and written informed consent was obtained from all of them either directly or from their parents (in the case of children <18 years old). The study was approved by the Ethics Committee of Universidad Simón Bolívar at Barranquilla, Colombia (approval # 00032, October 13, 2011).

### 2.2. Clinical Assessment

The *Grupo de Neurociencias del Caribe* carried out an extensive clinical, neurological, and neuropsychological evaluation to define ADHD status and the presence of other comorbidities, using a multi-stage scheme. The full evaluation protocol is described elsewhere [30]. Briefly, we employed the Diagnostic Interview for Children and Adolescents version IV (DICA-IV) [31,32] as the gold standard to assess the diagnosis of ADHD and/or ADHD-related comorbidities including CD and ODD in children, adolescents, and adults. For children and adolescents, the DICA-IV structured interview was completed by their parents who reported children’s symptoms and consequences in the academic, legal, and work-related areas, as well as alcohol and tobacco consumption, and its consequences [31,32,33]. This information was subsequently used to define the index case (proband). Presumptive ADHD diagnosis in childhood was assessed by obtaining a self-report retrospectively reporting on parents’ behaviour during grades 1 to 11 using the DICA-IV [34]. Persistent symptoms impacting family, social, and work-related environments were also recorded. Following the C criteria of DSM-IV, ADHD symptoms in children and adolescents were evaluated by their parents and teachers using the Colombian version of the Behavioural Assessment System for Children (BASC) [35] and the ADHD checklist [36,37]. Initially, 124 nuclear families with at least one child affected with ADHD were sequentially ascertained from patients attending a research program in ADHD [38], advertised in the *Grupo Neurociencias del Caribe’s* website.

### 2.3. SNPs Selection, DNA Extraction, and Genotyping

Twenty-six SNPs were selected from the ADHD gene data base (http://adhd.psych.ac.cn/) [39] as well as from previously reported associations to ADHD, ADHD endophenotypes, SUD, or ADHD comorbidities [2,3,6,18,40]. The full list of SNPs genotyped in this cohort is presented in Appendix A.

Genomic DNA was isolated from blood samples using the MasterPure^®^DNA Purification Kit (Epicentre Biotechnologies, Chicago, IL, USA) according to the manufacturer’s protocol and stored at −80 °C. DNA concentrations were measured using a NanoDrop spectrophotometer (Thermo Fisher Scientific, Waltham, MA, USA). The integrity of DNA samples was verified via 260/280 absorbance ratios. Genotyping was performed using multiplex Sequenom^®^ Technology on the Agena Bioscience MassARRAY^®^ MALDI-TOF instrument at the University of Arizona Genetics Core (UAGC). Primers designed for PCR amplification and extension were prepared by the UAGC. To avoid potential biases, samples were placed randomly, and laboratory personnel were blinded to the identity and source of DNA samples.

## 3. Statistical Analysis

### 3.1. Quality Control

SNPs with a minor allele frequency (MAF) ≥ 0.05 were classified as common and as rare otherwise [41]. Rare SNPs were excluded from the analysis. Additional parameters for excluding genetic markers from the analyses included (i) deviations from Hardy–Weinberg equilibrium with *p*-value < 0.05/*m*, where *m* is the number of SNPs being tested (26 in total, see Appendix A); (ii) a minimum genotype call rate of 80% [42,43,44,45]; (iii) the presence of more than two alleles; and (iv) MAF < 0.05. Allele and genotype frequencies were estimated using maximum likelihood. Mendelian errors, a common feature in SNP-based genotyping, were detected and subsequently corrected with the methods available in Golden Helix’s^®^ SNP variation suite (SVS) 8.4.0 (Golden Helix, Inc. Bozeman, MT, USA).

### 3.2. Genetic- and Haplotype-Based Association Analyses

Given that this cohort is comprised by nuclear families of size four (*n* = 33; 29.2%), five (*n* = 4; 3.5%) and six (*n* = 2; 1.8%), and not only by trios (*n* = 74; 65.4%), we used the family-based association test (FBAT) to study the association of SNPs and ADHD. The FBAT provides a unified framework to generalize the transmission disequilibrium test (TDT) [46,47], initially proposed to disclose genetic associations based on trios information. The FBAT accounts for different genetic models, sampling of family-based ascertainment designs, disease phenotypes, missing parents, and different null hypotheses [46]. The FBAT, as implemented in the PBAT module of SVS 8.4.0, allows testing combination of phenotypes (as a group) and genotypes that have the highest power by those predicted from the parents’ genotypes. As age and sex are known to impact ADHD susceptibility [48,49,50,51], both variables were included as covariates under the hypothesis of no linkage and no association. Adding these covariates substantially increases FBAT power [52,53]. Additive, dominant, recessive, and heterozygous advantage models of inheritance were explored. When testing the association between a particular biallelic marker and ADHD, using both the dominant and recessive genetic models of inheritance is equivalent to testing either of them.

The PBAT module also performs the FBAT and haplotype tests for selected combinations of phenotypes and markers on the actual patients’ genotypes, both as a group and individually, automatically controlling the Type I error rate to adjust for multiple comparisons [54], and the problem of population stratification that can lead to spurious associations [46,55,56,57,58]. FBAT screening methods are minimally affected by non-causal SNPs, since the final decision is based on the FBAT statistic [59].

FBATs use affected subjects as cases, and family members, parents, or siblings as “controls” (referred to as “unaffected individuals” from now on). Furthermore, low genotype call rates are compensated by the existence of parents’ genotypes, and paternity and Mendelian inconsistencies are also controlled [52,53]. All these features were essential to analyze our cohort, which includes complex family structures with multiple affected individuals and, in some cases, several probands, which introduces complex patterns of ascertainment. For interpretation purposes, the sign of the *p*-value of the FBAT indicates the direction of the effect; a positive *p*-value indicates susceptibility to ADHD, while a negative *p*-value indicates a protective effect.

Haplotype-based association analyses were also performed using the Parent TDT algorithm available in HaploView [60] including only SNPs located in the same chromosomal region.

## 4. Results

### 4.1. Family-Based Association Tests

Out of the 26 SNPs genotyped, 14 had one or more than two alleles, six had a MAF < 0.05, and six passed filters and quality control (Table 2). The total genotyping rate was 87.9% in all samples and 88.1% in the set of markers finally included for genetic analysis (Table 2).

We found that markers *SNAP25*-rs362990, *FGF1*-rs2282794, and *ADGRL3*-rs2122642 reached statistically significant FBAT statistics and confer susceptibility to ADHD (Table 3a). In particular, the T allele of marker *SNAP25*-rs362990 was found to confer susceptibility to ADHD under two different inheritance models (additive model, *p* = 2.46 × 10^−4^; heterozygous advantage (HA) model, *p* = 5.21 × 10^−4^; Table 3a). Furthermore, the two alleles of marker *FGF1*-rs2282794 confer susceptibility to ADHD in our cohort under three different inheritance models (A allele, dominant model *p* = 0.013; G allele, recessive model, *p* = 0.013; G allele, HA model, *p* = 0.016; Table 3a). Finally, the C and T alleles of marker *ADGRL3*-rs2122642 confer susceptibility to ADHD under a recessive (*p* = 0.035; Table 3a) or dominant (*p* = 0.035; Table 3a) model of inheritance, respectively.

### 4.2. Haplotype Block within ADGRL3 Confer Susceptibility to ADHD

We identified a haplotype block within the *ADGRL3* gene that confers susceptibility to ADHD (*p*_permuted_ < 0.05, Table 3b); this block spans 189 kb and is comprised by markers rs1565902, rs10001410, and rs2122642 (Figure 1). This CCC haplotype has a frequency of 41.1% in the full sample and is ~1.7 times more likely to be present in ADHD affected individuals than in unaffected individuals from these nuclear families (Table 3b).

## 5. Discussion

Family-based designs are robust against population admixture and stratification, and allow conducting complex segregation analysis and linkage and association studies that would be next to impossible in case/control-based designs [32,61,62,63]. With few exceptions [64,65,66,67], recent genetic studies on ADHD have primarily focused on case/control-based designs to study genetic contributions to ADHD susceptibility [15,16,18,68,69,70].

Here we evaluated the association between ADHD and intronic SNPs harboured in, among others, the *SNAP25*, *ADGRL3*, *FGF1*, *DRD4*, and *SLC6A2* genes (Appendix A) in 113 nuclear families ascertained from the metropolitan area of Barranquilla, Colombia. Despite that some of the genotyped markers have been reported as associated to ADHD [71], ADHD comorbidities [72,73], and ADHD endophenotypes [40], most genetic variants associated to ADHD have mainly been identified in populations with no African American background [14,15,74,75,76,77].

We found that markers rs362990-*SNAP25,* rs2282794-*FGF1*, and rs2122642-*ADGRL3* were also associated with ADHD in this set of nuclear families from the Colombian Caribbean coast (Table 3a). Moreover, we found that haplotype CCC (markers rs1565902-rs10001410-rs2122642, *p* = 0.021) within the *ADGRL3* gene confers susceptibility to ADHD in our set of nuclear families (Table 3b).

*SNAP25*, previously associated with ADHD and reduced expression in the prefrontal cortex [73], encodes a protein essential for synaptic vesicle fusion and neurotransmitter release and may play an important role in the synaptic function of specific neuronal systems [78]. Mutations within *SNAP25* may alter the level or function of the protein and hence may have an effect on the functions of synaptic vesicle fusion and neurotransmitter release [79]. Mouse models with a deletion of *SNAP25* show a hyperactive phenotype similar to ADHD in humans [80]. Although the same SNP was not genotyped in our sample, our genetic association result between a SNP harboured in *SNAP25* and ADHD is consistent with a case/control study in another Colombian sample with no African American component [81], which reported that individual SNPs and a haplotype within *SNAP25* were also associated with ADHD.

It is notable that SNPs in *FGF1* and *ADGLR3*, previously reported to be associated with ADHD endophenotypes and ADHD susceptibility in the Paisa genetic isolate [6,40], were also associated with ADHD in this set of nuclear families from the Colombian Caribbean coast. *FGF1* maps to 5q31.3, encodes a protein in the fibroblast growth factor (FGF) family and is expressed in key areas of the brain related to attention and activity (i.e., frontal cortex and the hippocampus) and to major depression (i.e., prefrontal cortex and the anterior cingulate cortex) [82] that may also be relevant to ADHD. The FGF family is involved in several cell survival activities including embryonic development, cell growth, morphogenesis, tissue regeneration, and tumour growth and invasion [83]. The encoded protein functions as a modifier of endothelial cell migration and proliferation, as well as an angiogenic factor, and has an important role in neural survival in Alzheimer’s disease [84,85,86]. The fact that marker rs2282794-*FGF1* is associated with ADHD in this Caribbean community reinforces the importance of further studying the *FGF1* gene as a novel candidate gene for ADHD [40].

*ADGRL3* is a member the latrophilin subfamily of G protein-coupled receptors and has already been implicated in ADHD susceptibility, predicting ADHD severity, disruptive behaviours comorbidity, long-term outcome, response to treatment, and SUD [2,3,6,14,15,16,17,18,19,20,21,74]. Animal models of ADHD have also provided convincing evidence of the critical role of *ADGRL3* in shaping the hyperactive/impulsive phenotype [87,88,89,90]. Besides further replicating the genetic contribution of these genes to ADHD, these results also suggest that these genetic effects are not unique to the Paisa community. Subsequent studies are needed to better understand how population stratification and different ethnic backgrounds, particularly African and Syrian–Lebanese, may impact ADHD susceptibility.

Altogether, our results suggest an association between SNP located in the *SNAP25*, *FGF1*, and *ADGRL3* genes and ADHD in this Caribbean community, which exhibits a strong genetic admixture between Aboriginal Amerindian communities with Spaniards and Africans, and later with other communities [22]. Although some clinical studies have been performed to better understand the manifestations of ADHD in individuals with an African American background [91,92,93,94], our study is the first to show that variants harbored in previously reported ADHD genes confer susceptibility to this disorder in such a population. Interestingly, this association is present when different genetic inheritance models are used (Table 3a). Future studies will include conducting complex segregation analyses to determine the inheritance mode of transmission (i.e., major gene, multifactorial contributors, or cohort effect) [10] of ADHD and comorbidities in this set of 113 families, the definition of cognitive and neuropsychological endophenotypes [40,95,96], and to perform linkage and association genetic analysis between common, rare, and functional exomic variants to ADHD, ADHD comorbidities [10,18,97], and reaction times [98]. In the future, we plan to perform genetic association analyses of the already genotyped SNPs (Appendix A) and ADHD comorbidities in this cohort, as well as in individuals with extreme ADHD phenotypes [3,10,29,99,100,101,102]. Further functional studies and possibly deep sequencing of these genes in this set of families could allow the identification of causal variants, and enhance translational medicine approaches to increase the accuracy of ADHD diagnosis and improve long-term outcomes.

## Figures and Tables

**Figure 1 cells-08-00907-f001:**
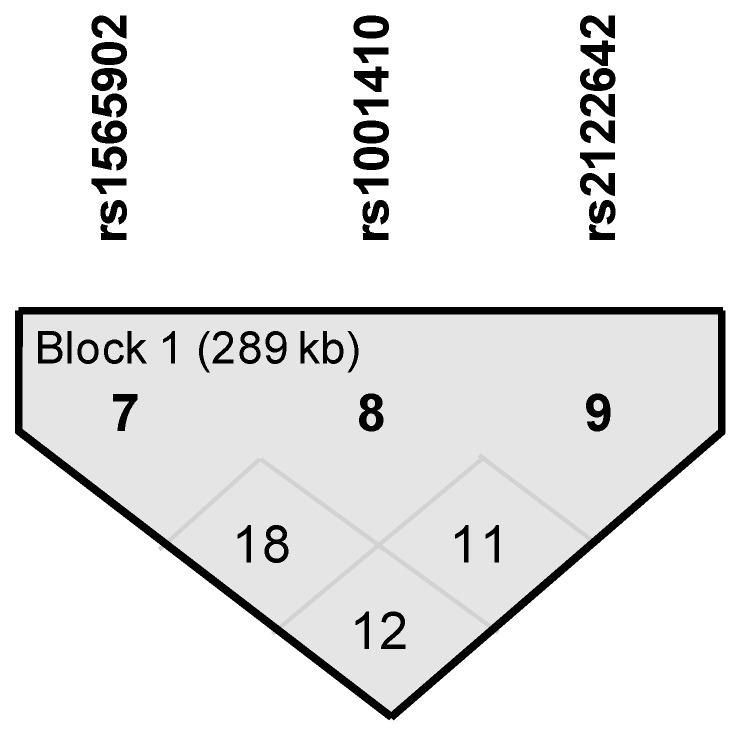
Linkage disequilibrium blocks for haplotype-based association analyses in 113 nuclear families segregating ADHD from a Caribbean community. See Table 3b for more information.

**Table 1 cells-08-00907-t001:** Demographic characteristics of individuals included in this study. Unaffected individuals were also ascertained from the 113 nuclear families but are clinically undiagnosed with ADHD.

	Affected	Unaffected	Statistic Index	*p*-Value	Effect Size
	*n* = 221	*n* = 165			
*Gender*	**Frequency (%)**	**Frequency (%)**	***χ*^2^**		
Male	151 (68.32)	70 (42.42)	24.849	<0.00001	—
Female	70 (31.68)	95 (57.58)			
	**Mean (SD)**	**Mean (SD)**	**Mann–Whitney’s *U***		
Age	21.4 (15.31)	33.9 (12.69)	26435	<0.0001	0.883

**Table 2 cells-08-00907-t002:** Statistics for genotyped markers passing quality control in 386 individuals segregating ADHD and belonging to 113 nuclear families from a Caribbean community.

Chr	Marker	Position *^a^*	Gene	Marker Information			
				Alleles *^b^*	MAF *^c^*	HW *p*-Value	%Genotyping
4	rs1565902	61,542,902	*ADGRL3*	C/**T**	0.473	0.048	89.4
4	rs10001410	61,608,511	*ADGRL3*	C/**T**	0.372	0.841	92.0
4	rs2122642	61,832,546	*ADGRL3*	C/**T**	0.330	0.326	91.5
5	rs2282794	142,602,144	*FGF1*	G/**A**	0.458	0.244	85.8
11	rs916457	637,014	*DRD4*	C/**A**	0.074	0.546	88.1
20	rs362990	10,295,573	*SNAP25*	T/**G**	0.120	0.788	80.3

*^a^* UCSC GRCh37/hg19 coordinates. *^b^* Minor allele reported in bold. *^c^* Sample-based estimate. Chr: Chromosome; MAF: Minor allele frequency; HW: Hardy–Weinberg. Additional details on these intronic markers are provided in Appendix A. Note: The *p*-value for the HW disequilibrium test was calculated for the full sample.

**Table cells-08-00907-t003a:** (a)

Chr	Marker	Gene	Position *^a^*	Marker Information			FBAT Results				
				Ref.	Observed	(Counts) [Frequency]	Allele	Cohort Frequency	Model	NIF	*p*-Value
20	rs362990	*SNAP25*	10,276,221	A	A/T	(4145/863) [0.828/0.172]	T	0.094	Additive	55	2.46 × 10^−4^
									HA	55	5.21 × 10^−4^
5	rs2282794	*FGF1*	141,981,709	G	A/G	(520/4488) [0.104/0.896]	A	0.458	Dominant	44	0.013
							G	0.542	Recessive	44	0.013
									HA	64	0.016
4	rs2122642	*ADGRL3*	62,698,264	G	C/T	(2722/2286) [0.543/0.457]	C	0.744	Recessive	45	0.035
							T	0.256	Dominant	45	0.035

**Table cells-08-00907-t003b:** (b)

Markers	Haplotype	Frequency	OR (T:U)	χ^2^	*p*-Value	
					Raw	Permuted
rs1565902-rs10001410-rs2122642	CCC	0.411	1.74 (74.1:42.5)	8.5	0.004	0.021

^a^ UCSC GRCh37/hg19 coordinates. Chr: Chromosome; HA: Heterozygous advantage; NIF: Number of informative families; FBAT: Family-based association test; OR: Odds ratio; T: Transmitted; UT: Untransmitted. Permuted *p*-values were obtained using 10,000 permutations as implemented in the HaploView’s ParentTDT algorithm [60]. All markers are intronic. Additional details on these markers are provided in Appendix A.

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
