# Peer review of "Genetic Variation Underpinning ADHD Risk in a Caribbean Community"

_cells, 2019, doi:10.3390/cells8080907_

Round 1

Reviewer 1 Report

Summary: The manuscript describes genetic association study between gene variants of ADGRL3 (LPHN3), SNAP25, FGF1, DRD4, and SLC6A2 genes are and the Attention-Deficit/Hyperactivity Disorder (ADHD) in the cohort of a total number of 113 Caribbean nuclear families consisted of n=386 participants. Family-based association studies suggested the role of rs362990-SNAP25, rs2282794-FGF1 and rs2122642-ADGRL3 variants in risk of ADHD. Additionally, haplotypes of ADGRL3 showed association as well in the Caribbean population, characterized by strong, African American, genetic component.

Specific comments and suggestions need to be addressed in order to remove a manuscript:

Abstract:

1)    “single nucleotide variants (SNVs)” – does it mean SNPs (single nucleotide polymorphisms)? Was there any reason not to use name SNP? Those variants can be found in SNP databases as dbSNP or Haploreg.

2)    “Afro-American component” – on the population level, the population is called African Americans.

3)    “The knowledge of population genetics is crucial to define risk and predict susceptibility to disease.” – this is a true and crucial statement, making me wonder if study provided suggestions of associations between different variants (or opposite alleles) when compared to other populations, e.g. Europeans.

Introduction:

1)    “In particular, common single nucleotide variants (SNVs) harbored in the Adhesion G-protein coupled receptor L3 (ADGRL3 also known as Latrophilin 3 or LPHN3)” – providing names of particular variants (with rs numbers) will be beneficial for future readers.   

Materials/Methods:

1)    “minimum genotype call rate of 80%” – why such low threshold was used instead of 90% or 95%?

2)    “monoallelic markers” – MAF will sort out monoallelic markers, as MAF will be 0.

3)    “we used the family-based association test (FBAT) to study the association of SNVs and ADHD. The FABT provides a unified framework to generalize the transmission disequilibrium test (TDT)" – typo is present in the second sentence, FABT needs to be changed into FBAT.

4)    “As age and sex are known to impact ADHD susceptibility, both variables were included as covariates under the hypothesis of no linkage and no association” – I would like to praise Authors for explaining why to include age and sex, literature citation will be a great asset.

5)    “Additive, dominant, recessive and heterozygous advantage models of inheritance were explored.” – exploration of genetic models will provide additional information, unfortunately, will cause issues with multiple testing as well. Also, ADHD is a complex phenotype and as such, the additive model can be applied.

6)    “automatically controlling the Type I error rate to adjust for multiple comparisons” – it reads to me as if the method is controlling for the number of variations analyzed, but not for all number of tests (e.g. different genetic models).

Results:

1)    I suggest adding only variants that passed the genetic test to Table 2. Also, scientific notation for p-values should be used only if needed, i.e. 3.3x10-1 is not readable and needs to be defined as 0.33. Lastly, I am not sure if the genetic position of an SNP is needed, I will rather replace it with localization (e.g. intronic). Such information will provide insights about potential, the biological role of the variation.  The same can be applied for Table 3a.

2)    “In particular, the T allele of marker rs362990-SNAP25 was found to confer susceptibility to ADHD under two different inheritance models (additive model, P=2.46x10-4; heterozygous advantage [HA] model, P=5.21x10-4; Table 3a).” – effect sizes should be reported near p-values. Based on p-values, the direction of effect cannot be assessed. I also wonder about the biological explanation about associations under different genetic models.

3)    Figure 1 is of poor quality. For most journals, 300 dpi is required.

4)    Table 3b shows the results of haplotype associations. I wonder why only 3 SNPs haplotypes were analyzed (others were not significant)? 1-2 sentences regarding haplotype analyses can be added to the “Methods” section.

Discussion:

1)    “also confers susceptibility to ADHD in our cohort” – p-value and effect size will make the discussion more lucid when compared to table references.

2)    The discussion is somewhat confusing: all three SNPs rs362990-SNAP25, rs2282794-FGF1, and rs2122642-ADGRL3 were previously reported in the literature. I propose to re-organize discussion as follows: 1) association found; 2) the previous report regarding SNP and 3) previous reports regarding gene.

3)    “In the meantime, we plan to perform genetic association analyses of the already” – in the meantime sounds not serious, a more suitable sentence is “In the future we plan to..”

4)    This study is very important, as it investigates understudied, Caribbean population. I suggest having a section comparing results from the study with results from mostly studied European population and to conclude whether results from Europeans can or cannot be generalized to other ethnic groups, 

Author Response

1) “single nucleotide variants (SNVs)” – does it mean SNPs (single nucleotide polymorphisms)? Was there any reason not to use name SNP? Those variants can be found in SNP databases as dbSNP or Haploreg.

Following your argument, we have change “SNV” by “SNP” in the revised version of the manuscript.

2) “Afro-American component” – on the population level, the population is called African Americans.

Corrected.

3) “The knowledge of population genetics is crucial to define risk and predict susceptibility to disease.” – this is a true and crucial statement, making me wonder if the study provided suggestions of associations between different variants (or opposite alleles) when compared to other populations, e.g. Europeans.

We appreciate your comment.

This is a good point. Every study performed in European communities has been consistent in showing association to the same allele/haplotype (Arcos-Burgos et al., 2019; Bruxel et al., 2015; Choudhry et al., 2012; Gomez-Sanchez et al., 2016; Kappel et al., 2017; Kappel et al., 2019; Hwang et al., 2015). Additionally, to the best of our knowledge, none of the markers reported in our study confer susceptibility to ADHD in populations with an African American component. However, markers rs10001410, rs1510921, rs2122642, rs6551678 and rs2282794 have been reported to be associated to ADHD endophenotypes in the Paisa genetic isolate (Mastronardi et al. 2016). Furthermore, marker rs916457 was associated with ADHD in a family-based sample of Caucasian individuals (Lasky-Su et al. 2007), and rs362990 was reported as associated with ADHD and OCD (Brem et al. 2014; Hawi et al. 2013).

References

1.     Arcos-Burgos, M., et al. (2019). "ADGRL3 (LPHN3) variants predict substance use disorder." Transl Psychiatry 9(1): 42.

2.     Brem et al. The neurobiological link between OCD and ADHD. Atten Defic Hyperact Disord. 2014 Sep;6(3):175-202. doi: 10.1007/s12402-014-0146-x

3.     Bruxel, E. M., et al. (2015). "LPHN3 and attention-deficit/hyperactivity disorder: a susceptibility and pharmacogenetic study." Genes Brain Behav 14(5): 419-427.

4.     Choudhry, Z., et al. (2012). "LPHN3 and attention-deficit/hyperactivity disorder: interaction with maternal stress during pregnancy." J Child Psychol Psychiatry 53(8): 892-902.

5.     Gomez-Sanchez, C. I., et al. (2016). "Attention deficit hyperactivity disorder: genetic association study in a cohort of Spanish children." Behav Brain Funct 12(1): 2.

6.     Hawi et al.  DNA variation in the SNAP25 gene confers risk to ADHD and is associated with reduced expression in prefrontal cortex. PLoS One. 2013 Apr 12;8(4):e60274. doi: 10.1371/journal.pone.0060274. 

7.     Hwang, I. W., et al. (2015). "Association of LPHN3 rs6551665 A/G polymorphism with attention deficit and hyperactivity disorder in Korean children." Gene 566(1): 68-73.

8.     Kappel, D. B., et al. (2017). "Further replication of the synergistic interaction between LPHN3 and the NTAD gene cluster on ADHD and its clinical course throughout adulthood." Prog Neuropsychopharmacol Biol Psychiatry 79(Pt B): 120-127.

9.     Kappel, D. B., et al. (2019). "ADGRL3 rs6551665 as a Common Vulnerability Factor Underlying Attention-Deficit/Hyperactivity Disorder and Autism Spectrum Disorder." Neuromolecular Med 21(1): 60-67.

10.  Lasky-Su  et al. Partial replication of a DRD4 association in ADHD individuals using a statistically derived quantitative trait for ADHD in a family-based association test. Biol Psychiatry. 2007 Nov 1;62(9):985-90.

11.  Mastronardi et al. Linkage and association analysis of ADHD endophenotypes in extended and multigenerational pedigrees from a genetic isolate. Mol Psychiatry. 2016 Oct;21(10):1434-40. doi: 10.1038/mp.2015.172. 

4) “In particular, common single nucleotide variants (SNVs) harbored in the Adhesion G-protein coupled receptor L3 (ADGRL3 also known as Latrophilin 3 or LPHN3)” – providing names of particular variants (with rs numbers) will be beneficial for future readers.

Corrected.

5) “minimum genotype call rate of 80%” – why such low threshold was used instead of 90% or 95%?

We appreciate your comment.

While we understand the reviewer’s concern that this threshold may have been a little low, this figure has been used by our group and other groups when exploring genetic associations in new or understudied communities, with a genotype call rate of 80% being the suggested threshold (Easton et al., 2007; Fu et al., 2009; Hunter et al., 2007; Whittaker et al., 2005; Jacob et al. 2014).

Another point of evidence of using such threshold for the genotype call rate is that in FBATs (kindly see the Methods section of the revised version of the manuscript), low genotype call rates “… are compensated by the existence of parents’ genotypes, and paternity and Mendelian inconsistencies are also controlled [50,51].”

References

1.     Easton et al. Genome-wide association study identifies novel breast cancer susceptibility loci. Nature 2007, 447, 1087-1093.

2.     Fu et al. Missing call bias in high-throughput genotyping. BMC Genomics 2009, 10, 106.

3.     Hunter et al. A genome-wide association study identifies alleles in FGFR2 associated with risk of sporadic postmenopausal breast cancer. Nat Genet 2007, 39, 870-874.

4.     Whittaker et al. SNP Analysis by MALDI-TOF Mass Spectrometry. In Cell Biology: A Laboratory Handbook, 3rd ed.; Celis, J., Simons, K., Small, J., Hunter, T., Shotton, D., Eds. Elsevier: Amsterdam, 2005.

5.     Jacob et al. A microarray platform and novel SNP calling algorithm to evaluate Plasmodium falciparum field samples of low DNA quantity. BMC Genomics. 2014 Aug 26;15:719.

6) “monoallelic markers” – MAF will sort out monoallelic markers, as MAF will be 0.

Corrected.

7) “we used the family-based association test (FBAT) to study the association of SNVs and ADHD. The FABT provides a unified framework to generalize the transmission disequilibrium test (TDT)" – typo is present in the second sentence, FABT needs to be changed into FBAT.

Done.

8) “As age and sex are known to impact ADHD susceptibility, both variables were included as covariates under the hypothesis of no linkage and no association” – I would like to praise Authors for explaining why to include age and sex, literature citation will be a great asset.

As suggested, we have included the following references to support this statement:

References

1.     Mowlem, F. D., et al. (2018). "Sex differences in predicting ADHD clinical diagnosis and pharmacological treatment." Eur Child Adolesc Psychiatry.

2.     Oerbeck, B., et al. (2019). "Adult ADHD Symptoms and Satisfaction With Life: Does Age and Sex Matter?" J Atten Disord 23(1): 3-11.

3.     Ramtekkar, U. P., et al. (2010). "Sex and age differences in attention-deficit/hyperactivity disorder symptoms and diagnoses: implications for DSM-V and ICD-11." J Am Acad Child Adolesc Psychiatry 49(3): 217-228 e211-213.

4.     Skogli, E. W., et al. (2013). "ADHD in girls and boys--gender differences in co-existing symptoms and executive function measures." BMC Psychiatry 13: 298.

9) “Additive, dominant, recessive and heterozygous advantage models of inheritance were explored.” – exploration of genetic models will provide additional information, unfortunately, will cause issues with multiple testing as well. Also, ADHD is a complex phenotype and as such, the additive model can be applied.

Thank you for your comment. As mentioned in the Methods section:

The PBAT module also performs the FBAT and haplotype tests for the selected combinations of phenotypes and markers on the actual patients’ genotypes, both as a group and individually, automatically controlling the Type I error rate to adjust for multiple comparisons [40,48-51].

In addition to the additive model, we also explored other models of genetic transmission due to the uniqueness of the population under study. As reported in the Results section (Table 3a), the additive model identified marker rs362990-SNAP25 to be statistically significantly associated with ADHD. However, given that some genetic studies support the dominant model of genetic transmission in ADHD (Arcos-Burgos et al, 2002; Arcos-Burgos et al, 2004; Wallis et al, 2008; Amin et al, 2009; Mastronardi et al, 2016; Corominas et al, 2018), we also tested this model in our sample. Our results indicate that markers rs2282794-FGF1 and rs2122642-ADGRL3 are associated to ADHD under this inheritance model (Table 3a).

There is an important point  regarding the tested inheritance genetic models we did not actually mention. When for a particular biallelelic marker one of the alleles (i.e., allele C) is found to be associated, the complementary allele (i.e., allele T) is also associated. The difference, however, is the frequency of each allele after performing the FBAT (kindly see Table 3a); while the frequency of allele C is pC, the frequency of allele T will be 1- pT. As a consequence, testing the Dominant or Recessive model of inheritance is equivalent to testing one of them, thus reducing the number of statistical tests being performed by 50%. This has now been clarified in the revised version of the manuscript.

References

1.     Arcos-Burgos et al., Attention-deficit/hyperactivity disorder (ADHD): feasibility of linkage analysis in a genetic isolate using extended and multigenerational pedigrees. Clin Genet. 2002 May;61(5):335-43.

2.     Arcos-Burgos et al., Pedigree disequilibrium test (PDT) replicates association and linkage between DRD4 and ADHD in multigenerational and extended pedigrees from a genetic isolate. Mol Psychiatry. 2004 Mar;9(3):252-9.

3.     Wallis et al.,  Review: Genetics of attention deficit/hyperactivity disorder. J Pediatr Psychol. 2008 Nov-Dec;33(10):1085-99. doi: 10.1093/jpepsy/jsn049. Epub 2008 Jun 3.

4.     Amin et al., Suggestive linkage of ADHD to chromosome 18q22 in a young genetically isolated Dutch population. Eur J Hum Genet. 2009 Jul;17(7):958-66. doi: 10.1038/ejhg.2008.260.

5.     Mastronardi et al., Linkage and association analysis of ADHD endophenotypes in extended and multigenerational pedigrees from a genetic isolate. Mol Psychiatry. 2016 Oct;21(10):1434-40. doi: 10.1038/mp.2015.172. 

6.     Corominas et al.,  Identification of ADHD risk genes in extended pedigrees by combining linkage analysis and whole-exome sequencing. Mol Psychiatry. 2018 Aug 16. doi: 10.1038/s41380-018-0210-6.

10) “automatically controlling the Type I error rate to adjust for multiple comparisons” – it reads to me as if the method is controlling for the number of variations analyzed, but not for all number of tests (e.g. different genetic models).

FBAT allows to control the type I error rates and hence overcome the multiple comparison problem when analyzing genetic data. This is possible because FBAT screening methods are minimally affected by the non-causal SNPs, and are robust against effects of population stratification and admixture (see Methods section). The relevant part now reads:

“When testing the association between a particular biallelic marker and ADHD, using both the dominant and recessive genetic models of inheritance is equivalent to testing either of them. Thus, the number of statistical tests being performed is reduced in half.

The PBAT module also performs the FBAT and haplotype tests for the selected combinations of phenotypes and markers on the actual patients’ genotypes, both as a group and individually, automatically controlling the Type I error rate to adjust for multiple comparisons[52], and the problem of population stratification that can lead to spurious associations [44,53-56].

FBAT screening methods are minimally affected by non-causal SNPs, and are robust against effects of population stratification and admixture, since the final decision is based on the FBAT statistic [57]. FBATs use affected subjects as cases, and family members, parents, or siblings as “controls” (referred to as “unaffected individuals from now on”). Furthermore, low genotype call rates are compensated by the existence of parents’ genotypes, and paternity and Mendelian inconsistencies are also controlled [50,51].”

Regarding testing different genetic models, kindly see our response to Comment #9.

11) I suggest adding only variants that passed the genetic test to Table 2. Also, scientific notation for P-values should be used only if needed, i.e. 3.3x10-1 is not readable and needs to be defined as 0.33. Lastly, I am not sure if the genetic position of an SNP is needed, I will rather replace it with localization (e.g. intronic). Such information will provide insights about potential, the biological role of the variation. The same can be applied for Table 3a.

Following your suggestion, we added a footnote specifying the localization of all SNPs in both Table 2 and Table 3a.

12) “In particular, the T allele of marker rs362990-SNAP25 was found to confer susceptibility to ADHD under two different inheritance models (additive model, P=2.46x10-4; heterozygous advantage [HA] model, P=5.21x10-4; Table 3a).” – effect sizes should be reported near P-values. Based on P-values, the direction of effect cannot be assessed. I also wonder about the biological explanation about associations under different genetic models.

As specified in Section 3.2.,

“For interpretation purposes, the sign of the P-value of the FBAT indicates the direction of the effect; a positive P-value indicates susceptibility to ADHD, while a negative P-value indicates a protective effect.”

It is a well-known that estimating effect sizes or disparity measures such as the Odds Ratio (OR) is not possible when using FBAT. That being said, our conclusions rely on the direction of P-values (see above and Methods section), which are calculated based on the FBAT test statistic.

13) Figure 1 is of poor quality. For most journals, 300 dpi is required.

Corrected.

14) Table 3b shows the results of haplotype associations. I wonder why only 3 SNPs haplotypes were analyzed (others were not significant)? 1-2 sentences regarding haplotype analyses can be added to the “Methods” section.

Thank you for your comment. As specified in the Statistical Analysis section, only SNPs within the same chromosomal region were included for haplotype-association analysis.

15) “also confers susceptibility to ADHD in our cohort” – P-value and effect size will make the discussion more lucid when compared to table references.

Kindly see our response to Comment #12.

16) The discussion is somewhat confusing: all three SNPs rs362990-SNAP25, rs2282794-FGF1, and rs2122642-ADGRL3 were previously reported in the literature. I propose to re-organize discussion as follows: 1) association found; 2) the previous report regarding SNPs and 3) previous reports regarding gene.

Thank you very much for your input. Following your suggestion, we have restructured the Discussion accordingly in the revised version of the manuscript.

17) “In the meantime, we plan to perform genetic association analyses of the already” – in the meantime sounds not serious, a more suitable sentence is “In the future we plan to..”

Corrected.

18) This study is very important, as it investigates understudied, Caribbean population. I suggest having a section comparing results from the study with results from mostly studied European population and to conclude whether results from Europeans can or cannot be generalized to other ethnic groups.

Following your suggestion, we have added a couple of lines in the Discussion in the revised version of the manuscript.

Reviewer 2 Report

Puente-Rozo et al., performed a family-based association study of previous SNPs associated with ADHD in a mixed population from Colombia. The manuscript is very well written, and it is very easy to understand. Reports of previously associated SNPs deserve considerations in populations other than Caucasians, which has been the focus of association studies thus far. Although, I think this study deserves publication, it has several methodological and conceptual issues:

1)      There is a substantial difference between SNP (single nucleotide polymorphism) and SNV (single nucleotide variant). The first (SNP) is used to describe the common variations in the genome (usually variants with MAF>0.01 in general population). The second (SNV) is used to describe the rare variants (MAF<0.01) which are usually detected with next-generation sequencing. The authors keep defining along the manuscript common polymorphisms as “SNVs” (i.e. “SNVs with a minor allele frequency (MAF) ≥ 0.01 were classified as common and as rare otherwise”), which is a conceptual mistake. Please correct the terminology changing SNV to SNP.

2)      It is confusing the description of your sample, given that this is a family–based design, there are not “controls” (i.e. line 190: “more likely to be present in ADHD affected individuals than in controls (Table 3b)”). I don’t understand if the authors refer in Table 1 and in “2.1. subjects” to unaffected as unaffected siblings or they refer to parents. Unaffected siblings are generally not used in the FBAT software statistics (unless is a discordant analysis), and I am not sure if these 165 individuals include only parents, or a mix between parents and unaffected sibs. Please amend this part.

3)      Table 1 and Table S1 are confusing. The authors should include in Table S1 all the 26 SNPs (sorted by chr/position/gene) and refer for each the QC process and the reason for the QC exclusion (which is monomorphic, which has 3 alleles, which has been excluded for HWE, etc). Include in this table S1 also the reported MAF from dbSNP or any other database. The Table 1 should include only the SNPs that passed QC (8 SNPs). It looks a bit weird that 70% of your SNPs are excluded in the QC process when these SNPs have been selected and reported significant in previous studies. Can the authors explain why? It looks that there is a problem in the experimental design and or genotyping.

4) The most concerning part are the statistical procedures. Only 8 SNPs are analysed with FBAT under 3 genetic models (recessive, dominant and additive). Why the authors do not include in Table 3 all the results for these 8 SNPs? Under these data the Bonferroni’s correction is P<0.00208 (0.05/24 (8-SNPsX3models). There is no reference to any statistical correction adopted for multiple testing. Given your data the only SNP significant after correction is rs362990-SNAP25. I am a supporter for the additive models in complex disorders, and I would use only this model to avoid the corrections for a consistent number of tests. Also there is a methodological problem with the haplotype analysis. Haplotype analyses make sense only when a SNP is significantly associated during a single-marker analysis, and should not be applied to all SNPs previously genotyped. Thus, because no SNP passed multiple testing in ADGRL3, haplotype analyses cannot be performed. Also the extreme significant P-value obtained for the block 2 (CAA, P-val<0.0001) is driven probably from the rs6551660 which is not in HWE (table 2), but that the authors include anyway in this analysis.

Author Response

1) There is a substantial difference between SNP (single nucleotide polymorphism) and SNV (single nucleotide variant). The first (SNP) is used to describe the common variations in the genome (usually variants with MAF>0.01 in general population). The second (SNV) is used to describe the rare variants (MAF<0.01) which are usually detected with next-generation sequencing. The authors keep defining along the manuscript common polymorphisms as “SNVs” (i.e. “SNVs with a minor allele frequency (MAF)³ 0.01 were classified as common and as rare otherwise’’), which is a conceptual mistake. Please correct the terminology changing SNV to SNP.

Done.

2) It is confusing the description of your sample, given that this is a family–based design, there are not “controls” (i.e. line 190: “more likely to be present in ADHD affected individuals than in controls (Table 3b)”). I don’t understand if the authors refer in Table 1 and in “2.1. subjects” to unaffected as unaffected siblings or they refer to parents. Unaffected siblings are generally not used in the FBAT software statistics (unless is a discordant analysis), and I am not sure if these 165 individuals include only parents, or a mix between parents and unaffected sibs. Please amend this part.

Done.

In all cases, we referred to individuals not diagnosed with ADHD from the set of 113 nuclear families when wrote “ADHD unaffected individuals” throughout the paper. However, referring to ADHD unaffected individuals as “controls” comes from the FBAT classical notation (see Methods section in the revised version of the manuscript, lines 181-183).

We have now clarified this in the caption of Table 1 by adding Unaffected individuals were also ascertained from the 113 nuclear families but are clinically undiagnosed with ADHD. We also changed line 190 accordingly.

3) Table 1 and Table S1 are confusing. The authors should include in Table S1 all the 26 SNPs (sorted by chr/position/gene) and refer for each the QC process and the reason for the QC exclusion (which is monomorphic, which has 3 alleles, which has been excluded for HWE, etc). Include in this table S1 also the reported MAF from dbSNP or any other database. The Table 1 should include only the SNPs that passed QC (8 SNPs). It looks a bit weird that 70% of your SNPs are excluded in the QC process when these SNPs have been selected and reported significant in previous studies. Can the authors explain why? It looks that there is a problem in the experimental design and or genotyping.

Thank you for your comment. As suggested, we have modified both tables in the revised version of the manuscript.

We perfectly understand the reviewer’s concern regarding genotyping problems. We followed all standard procedures to genotype our sample and hence can say that this phenomenon is not a reflection of problems in the experimental design. However, we acknowledge that genotyping problems may be occurring in this case, possibly due to the characteristics of our population. In particular, it is possible that SNPs already reported/genotyped in populations do not vary much in this Caribbean community and hence were excluded after QC assessment. Two examples are the intronic marker rs11568324-SLC6A22 and the exonic marker rs1800443-DRD4, both of which are monomorphic in our sample, but are vary slightly in other populations.

4) The most concerning part are the statistical procedures. Only 8 SNPs are analyzed with FBAT under 3 genetic models (recessive, dominant and additive). Why the authors do not include in Table 3 all the results for these 8 SNPs? Under these data the Bonferroni’s correction is P<0.00208 (0.05/24 (8-SNPsX3models). There is no reference to any statistical correction adopted for multiple testing. Given your data the only SNP significant after correction is rs362990-SNAP25. I am a supporter for the additive models in complex disorders, and I would use only this model to avoid the corrections for a consistent number of tests. Also there is a methodological problem with the haplotype analysis. Haplotype analyses make sense only when a SNP is significantly associated during a single-marker analysis, and should not be applied to all SNPs previously genotyped. Thus, because no SNP passed multiple testing in ADGRL3, haplotype analyses cannot be performed. Also the extreme significant P-value obtained for the block 2 (CAA, P <0.0001) is driven probably from the rs6551660 which is not in HWE (table 2), but that the authors include anyway in this analysis.

We appreciate your comments.

Results for all SNPs were not reported as they were not statistically significant.

Regarding correction for multiple testing and genetic models, kindly see our responses to Comments #9 and #10 of Reviewer #1. We made an unintentional mistake in the previous version of Table 2, which we have now been fixed in the revised version of the manuscript; marker rs6551660 does, in fact, pass HWE. We are sorry for the confusion.

Because of how FBAT deal with multiple testing (kindly see our response to Comment #10 of Reviewer #1), marker rs2122642-ADGRL3 passed multiple testing. Given previous association studies suggesting a potential role of ADGRL3 in ADHD susceptibility, we decided to explore potential association between ADHD and haplotypes within ADGLR3. Thus, markers passing QC harbored in these genes were considered for further analysis. Although we understand the reviewer’s concern as to why we included nonsignificant SNPs for haplotype analysis, it is possible to perform such analysis even if markers comprising the haplotype are not associated with ADHD at the individual level; this rational is similar to that in multipoint linkage analysis. Unlike SNP-based analysis, haplotype-based analysis may help improve the detection of causal genetic variants and combine information from multiple SNPs to identify such variants. In our case, we identified two highly frequent haplotypes within ADGRL3 that confer susceptibility to ADHD (Table 3b).

Reviewer 3 Report

 Puentes-Rozo et al carried out gene association study to identify 26 SNPs known to be related to ADHD. Overall, the manuscript is concise and well written. Although the study lacks novelty, the replicated study could be of substantial interest to study the genetic risk to ADHD. I have following comments:

a. The authors are encouraged to include statistical analysis section separately in the manuscript’s material & methods section and include how different variables were chosen & categorized.

b. The overall impact of this manuscript remains unclear. As the SNPs identified in this paper are already known to be related to ADHD in Colombian population (Line 222-224), the observation that the same SNPs are related to ADHD in Colombia (this study) does not reveal new insights to potential readers. Authors are required to add some sentences that describe the goal and impact of this study to enhance the enthusiasm of the manuscript.

Author Response

1) The authors are encouraged to include statistical analysis section separately in the manuscript’s material & methods section and include how different variables were chosen & categorized.

Thank you for your comment.

As suggested, we have now included a new Statistical Analysis section in the revised version of the manuscript.

Regarding how different variables were chosen and categorized, it is important to mention that other than categorizing age, which is routinely used in ADHD studies, we did not performed any type of categorization on other variables. The relevant part in the paper where we explain this procedure reads (subsection 2.1):

94 (26.2%) were children (6-11 years), 34 (9.4%) adolescents (12-17 years) and 232 (64.4%) adults (>18 years)”.

2) The overall impact of this manuscript remains unclear. As the SNPs identified in this paper are already known to be related to ADHD in Colombian population (line 222-224), the observation that the same SNPs are related to ADHD in Colombia (this study) does not reveal new insights to potential readers. Authors are required to add some sentences that describe the goal and impact of this study to enhance the enthusiasm of the manuscript.

The reviewer is right. However, it is important to point out that none of the markers reported in this study have previously been reported as ADHD associated in populations with an African American component (see also our response to Comment #3 of Reviewer #1). Thus, our study is the first showing that variants harbored in previously reported ADHD genes confer susceptibility to this disorder in such population (kindly see our response to Comment #5 of Reviewer #1).

Following your suggestion, we have added a couple of lines highlighting this special contribution of our study in the Discussion of the revised version of the manuscript.

Round 2

Reviewer 2 Report

The authors improved the quality and the reading of this paper, but there are several methodological issues that have not been assessed and the quality of the association study is somehow alarming. The authors in this revised version include now rs6551660 as one of the SNPs that passed QC, although the HWE is =0.0002 and threshold is indicated as (0.05/26 total SNPs=0.0019). Generally the HWE should be kept in small association studies (like this presented here) not so stringent to make sure about the quality of the genotyping (HWE=around 0.01). In this case the SNP included in the family-based association study rs6551660 is definitely not in HWE, and should be removed. I have not noticed in the previous version that the authors should also remove other SNPs that are not in HWE considering their threshold, such as rs1510921 (HWE=0.0014). This QC would as explained in the previous version change the haplotype analysis, because rs6551660 cannot be included in this analysis. Remarkably, the authors did not perform any test for multiple comparisons (under 3 genetic models), and the comment provided about the biallelic markers does not make sense form a statistical point of view.